# Update on Functional Inhibitors of Acid Sphingomyelinase (FIASMAs) in SARS-CoV-2 Infection

**DOI:** 10.3390/ph14070691

**Published:** 2021-07-18

**Authors:** Gwenolé Loas, Pascal Le Corre

**Affiliations:** 1Department of Psychiatry, Hôpital Erasme, Université Libre de Bruxelles (ULB), 1070 Brussels, Belgium; 2Research Unit (ULB 266), Hôpital Erasme, Université Libre de Bruxelles (ULB), 1050 Brussels, Belgium; 3Pôle Pharmacie, Service Hospitalo-Universitaire de Pharmacie, CHU de Rennes, 35033 Rennes, France; plcuniv@icloud.com; 4Irset (Institut de Recherche en Santé, Environnement et Travail)-Inserm UMR 1085, University of Rennes, CHU Rennes, INSERM, EHESP, 35000 Rennes, France; 5Laboratoire de Biopharmacie et Pharmacie Clinique, Faculté de Pharmacie, Université de Rennes 1, 35043 Rennes, France

**Keywords:** functional inhibitors of acid sphingomyelinase (FIASMAs), acid sphingomyelinase, COVID-19, SARS-CoV-2, mortality

## Abstract

The SARS-CoV-2 outbreak is characterized by the need of the search for curative drugs for treatment. In this paper, we present an *update of knowledge about* the interest of the functional inhibitors of acid sphingomyelinase (FIASMAs) in SARS-CoV-2 infection. Forty-nine FIASMAs have been suggested in the treatment of SARS-CoV-2 infection using in silico, in vitro or in vivo studies. Further studies using large-sized, randomized and double-blinded controlled clinical trials are needed to evaluate FIASMAs in SARS-CoV-2 infection as off-label therapy.

## 1. Introduction

The severe acute respiratory syndrome coronavirus 2 (SARS-CoV-2) is identified as the disease-causing pathogen of Coronavirus disease 2019 (COVID-19). Up to June 2021, the number of confirmed cases worldwide exceeded 179 million, with more than 3.9 million deaths. Some encouraging results have been observed using dexamethasone and remdesivir in the treatment of COVID-19 in patients requiring supplemental oxygen, mechanical ventilation or extracorporeal membrane oxygenation, but there is still no drug preventing host cell infection or cytokine release syndrome. 

SARS-CoV-2 is a family of enveloped viruses that enter into host cells by receptor binding and membrane fusion [1], and acid sphingomyelinase (ASM) and ceramide play a prominent role in receptor signaling and infection cycle [1]. There are two forms of ASM: (i) lysosomal ASM that is transported to the endosomal compartment and anchored to the inner lysosomal membrane, and (ii) secretory ASM that is transported to the outer leaflet of the plasma membrane.

Some organic molecules, including currently marketed drugs, have the potential to functionally inhibit the activity of ASM from reversible and additive manner. These molecules identified by the acronym FIASMAs (i.e., functional inhibitors of acid sphingomyelinase) have the potential to inhibit this enzyme, notably the lysosomal ASM, and therefore disrupt the entry of viruses into cells [2]. In cell culture models, inhibition of acid sphingomyelinase activity by amitriptyline has been shown to prevent the infection of cells with SARS-CoV-2 and pseudoviral SARS-CoV-2 in an ex vivo model [3]. Hence, the impact of chronic exposure of patients to a drug or a group of drugs with FIASMA properties on the clinical course of patients infected with SARS-CoV-2 may be questioned.

At least sixty-four drugs classified as FIASMAs reduce ASM activity by at least 50% at 10 µM concentration. The distribution of FIASMAs with respect to their ATC code revealed that specific therapeutic groups were over-represented: C08 (calcium channel blockers; amlodipine), D04 (antipruritics; promethazin), N04 (anti-Parkinson’s; benztropine), R06 (antihistamines for systemic use; astemizole), N06 (psychoanaleptics; fluvoxamine) and N05 (psycholeptics; chlorpromazine).

Repurposing of drugs in COVID-19 is being investigated using in silico, in vitro or ex vivo activity-based studies, as well on in vivo activity-based drug repurposing in animal models and human studies (epidemiological, clinical studies, case reports). 

Recently (9 January 2021), we reviewed [4] all the corresponding studies (N = 27), including the pre-print publications, and found that thirty-two FIASMAs could be considered as potential drugs for treating SARS-CoV-2. 

The aim of the present overview is firstly to present a brief history of the interest of FIASMAs in SARS-CoV-2 infection, and secondly, to provide an updated overview of studies on FIASMAs in this infection. 

## 2. Brief History of the Studies of FIASMAs in the SARS-CoV-2 Infection

In 2005, a study reported the role of increased ASM activity in organ failure of patients with severe sepsis [5].

In 2010, compounds including marketed drugs with a potential to inhibit ASM activity in vitro were called “Functional inhibitors of ASM” (FIASMAs). These compounds are cationic amphiphilic molecules with a relative heterogeneity in terms of chemical structure. They typically are polycyclic molecules, with at least 1 basic nitrogen atom (pKa > 4, which corresponds to a partially protonated functional group at acidic pH) and show moderate to high lipophilicity (log*p* > 3) [2].

In 2014, two different in vitro studies reported that several FIASMAs were active against both MERS-CoV and SARS-CoV [6,7]. In these two studies, inhibition of ASM was not mentioned, and other active mechanisms were suggested as inhibitors of clathrin-mediated endocytosis for chlorpromazine. 

In 2019, Andrews [8] suggested that “It is becoming increasingly clear that many pathogens that produce membrane damaging also trigger a repair mechanism involving exocytosis of lysosomal ASM, generating ceramide-enriched cell surface domains that facilitate cell invasion”.

Independently of the ASM pathway, viruses usually take advantage of the endocytosis mechanisms to penetrate the cytosol of cells, and different mechanisms of internalization are involved. These are clathrin-mediated endocytosis, macropinocytosis, caveolar/lipid raft-mediated endocytosis, as well as several less well-characterized clathrin- and caveolin/lipid raft-independent mechanisms [9].

Markus Blaess in a pre-print (May 5, 2020) [10] strongly suggested that lysosomotropic compounds could protect against COVID-19 infection in a concealed way, and cited a set of 34 clinically approved lysosomotropic compounds, 30 of them being FIASMAs. The author suggested that these compounds could be used off label using either local (inhalative) or systematic administration. Later, the author [11] developed the repurposing of lysosomotropic drugs (including numerous FIASMAs) in COVID-19 infection.

In 2020, among the 23 in silico, in vitro or ex vivo studies reporting activity of FIASMAs against SARS-CoV-2 [4], only two cited the inhibition of ASM as the principal mechanism of action of the drugs [3,12].

In 2021, Chung and Claus [13] reviewed the function of ASM to explore the question whether ASM is a friend or foe in the course of sepsis and severe infection (not limited to COVID-19 infection). The interest of FIASMAs repurposing and particularly amitriptyline were underlined by the authors.

In 2021, a study [14] used SARS-CoV-2 pseudoviruses to infect human angiotensin-converting enzyme 2 (ACE2)-expressing HEK293T cells and evaluated virus infection. SARS-CoV-2 entry was dependent on ACE2 and sensitive to pH endosome/lysosome in HEK293T cells. Moreover, the infection of SARS-CoV-2 pseudoviruses was independent of dynamin, clathrin, caveolin, endophilin as well as micropinocytosis. Cholesterol-rich lipids rafts and endosomal acidification are key steps of SARS-CoV-2 required for infection of host cells.

## 3. Update (16 June 2021) on Studies with FIASMAs in the SARS-CoV-2 Infection

To review the extensive evidence about FIASMAs as a therapeutic modality for COVID-19, authors attempted to answer the following key questions. First, would FIASMAs provide benefits in relation to COVID-19? Second, should FIASMAs be used as early intervention in COVID-19 disease?

We manually searched two electronic databases, PubMed and Google Scholar, for English-language titles and abstracts using the terms “Alverine OR… OR… Trimipramine” (64 drugs, see [2]) and “COVID-19 OR SARS-CoV-2”

Among the 231 articles (PubMed and Google Scholar), 91 were retained [3,12,15,16,17,18,19,20,21,22,23,24,25,26,27,28,29,30,31,32,33,34,35,36,37,38,39,40,41,42,43,44,45,46,47,48,49,50,51,52,53,54,55,56,57,58,59,60,61,62,63,64,65,66,67,68,69,70,71,72,73,74,75,76,77,78,79,80,81,82,83,84,85,86,87,88,89,90,91,92,93,94,95,96,97,98,99,100,101,102,103]. Twenty-seven have been reviewed in our previous article [4] (23 in silico, in vitro or in vivo studies, 5 human studies (one common with in vitro studies)) and 64 were new studies highlighting the interest of the scientific community for FIASMAs. Among these new studies, there were 52 in silico, in vitro or in vivo studies, and 14 were human studies (2 of them also reporting in silico or in vitro studies)

Forty-nine FIASMAs have been suggested as potential treatment in SARS-CoV-2 infection and nine of them (benztropine, chlorpromazine, clomipramine, emetine, fluphenazine, loperamide, promethazine, tamoxifene and triflupromazine) were active on the three coronaviruses (SARS, SARS-MERS, SARS-CoV-2) (see Table 1).

Fifteen FIASMAs [20,25,29,30,31,32,46,53,54,73,75,76,77,78,82,88,101,102,103] have been studied using epidemiological-clinical studies or case reports (amiodarone, amitriptyline, amlodipine, carvedilol, chlorpromazine, clomipramine, desloratadine, fluoxetine, fluvoxamine, hydroxyzine, loperamide, loratadine, melatonine, paroxetine, and sertraline). Among the 15 FIASMAs, only amlodipine [32] and fluvoxamine [75] were studied using randomized double-blind clinical studies (see Table 2).

Among the 19 human studies, there were two case reports on amiodarone [20] or loratadine [88], four retrospective studies that have explored the association between one FIASMAs (carvedilol [25,53], hydroxyzine [77,101], loratadine [77], melatonine [25,53], paroxetine [53]) and the negativity or positivity on a PCR test, four prospective studies on fluvoxamine [75,76], melatonine [102] or amlodipine [32] including two randomized clinical trial [32,75], 4 retrospective studies on mortality in hospitalized COVID-19 patients on amlodipine [29,30,31] or melatonine [82], one observational study on low rate of COVID-19 infection in psychiatric patients treated by antipsychotics comparatively to nurses or physicians [73], 4 retrospective studies on mortality or intubation on hospitalized COVID-19 patients on amiodarone, amitriptyline, amlodipine, chlorpromazine, clomipramine, desloratadine, fluoxetine, hydroxyzine, paroxetine and sertraline [46,54,78,103].

Several methodological aspects of the abovementioned studies deserve to be discussed. First, the choice of the primary end point can be questioned in some studies, especially those based on clinical deterioration (e.g., percentage of admission in ICU, time to intubation, or percentage of intubation …) or those using a composite criterion as opposed to those based only on mortality rate. Indeed, the former studies can display significant variability as a function of time (bed occupancy rate, number of other cases requiring intubation, pandemia severity …) and also according to hospitals, thus potentially leading to problems of reproducibility in the criteria in multicenter retrospective studies [46,54,78,103]. Hence, studies based on a mortality criterion should be considered more relevant. The second point to be considered is the choice of the FIASMAs and the duration of their prescription within the study. Indeed, this latter parameter influences the accumulation in pulmonary tissue, and more specifically in lysosomes, to achieve the desired inhibition of acid sphingomyelinase [11]. The highest accumulation in plasma and also into the lysosome compartments (i.e., called steady-state) requires around 7 times the apparent elimination half-life of the drug to occur. This can lead to a quite long delay depending on the drug (e.g., from 7 to 14 days for fluoxetine), especially for an infection where deterioration and mortality arise quite rapidly in patients with severe symptoms. Besides the time to reach the highest accumulation depending on the elimination half-life, the magnitude of the tissue distribution has to be considered. As lipophilic and amine drugs, most FIASMAs have a very high volume of distribution [4], so that they have the potential to distribute freely in some body organs. More precisely, some of these drugs have been called “pneumophilic drugs” as a result of their ability to distribute in the lungs (e.g., amiodarone, verapamil, tricyclic antidepressants like imipramine and amitriptyline, and phenothiazines like chlorpromazine, β-blockers like propranolol, and local anesthetics like lidocaine). The mechanism of this lung accumulation currently remains unclear but carrier-mediated pathways have been suggested. Even though their volume of distribution may be quite large, these drugs are also very highly bound to plasma proteins (>95%) so that drug-free levels in the body should be quite low. Hence, in order to estimate the potential effect of a drug on the course of COVID-19 infection in patients from retrospective studies, it would be necessary to make sure that the patients included in the study have previously been exposed to the drug(s) being studied at the time of inclusion, ideally by a chronic prescription. Given these pharmacokinetic features, in our previous paper [4] we have underlined that reaching a steady state to obtain a maximal effect would require a delay (seven times the half-life), and alternatively that a loading dose may be used to rapidly reach the steady state [11].

The direct administration through the pulmonary tract may help in reaching pulmonary targets, and should be an option to consider since COVID is a respiratory disease (11). Ambroxol, a lipophilic cationic molecule, not formally categorized as FIASMA, is approved as a mucolytic drug, and can be used by inhalation. In a recent ex vivo study, it has been shown that ambroxol reduced ASM activity in human nasal epithelial cells infected with pp-SARS-CoV-2 spike [100].

When examining the results of retrospective studies in COVID-19 hospitalized patients, it is important to keep in mind that the mean length of hospital stay is 13 days with around an 8-day delay from symptom onset to admission, and 3-day delay from admission to ICU (ISARIC clinical data report 4 October 2020 on 102,959 individuals from 566 sites across 42 countries).

In other terms, if the study baseline was defined as the date of hospital admission, and notably within the first 48 h, then the steady state is unlikely to be obtained for most drugs within the natural course of the disease except for the FIASMAs having a short half-life (melatonine: 7 half-lives: 6–7 h, alverine: 7 half-lives: 6 h, cloperastine: 7 half-lives: 24 h).

Among the different retrospective studies on COVID-19 hospitalized patients, two studies have included patients without mentioning the duration of the prescription of the drugs [29,30], one study explored the interest of melatonine [82], only one study has included patients with FIASMAs taken for at least 7 half-lives [31], and four studies have taken into account only FIASMAs prescribed after the hospitalization. More precisely, for these four studies, the beginning of the prescription was: the date of the first prescription (during hospitalization) of chlorpromazine [46] or hydroxyzine [78],—within the first 48 h of hospital admission for receiving any antidepressants [54],—within the first 24 h of hospital admission for receiving at least one FIASMAs [103]. However, among the clinical studies, negative results were reported for amlodipine, amiodarone, amitriptyline, clomipramine, chlorpromazine, desloratadine, fluoxetine, hydroxyzine, paroxetine, sertraline [32,46,54,103] (see Table 2). These negative results could be explained first by the acute drug prescription, second by underpowered studies (an insufficient power of the statistical tests) explained by low samples size, and third by selection biases.

Regarding the two case reports, those [20] reporting the use of amiodarone during 5 days, although the half-life of the drug is around 50 days, calls into question the FIASMA activity of the drug.

Regarding the prospective studies, and notably the randomized clinical studies using either fluvoxamine or amlodipine, the protocol required the drug intake after the randomization to patients having a positive RT-PCR.

For the randomized double-blind clinical study exploring the potential effect of fluvoxamine against placebo [75], the main criterion was clinical deterioration within 15 days so that the half-life of the drug (17–22 h, steady-state: 5–6 days) allowed to test for ASM activity.

For the randomized double-blind clinical study [32] exploring the potential effect of amlodipine against losartan, the authors had two principal criteria: 30-day mortality and length of hospital stay. They reported no significant difference on the two criteria, but in their study, a stratified randomization of age has not been done, and the mean age of the groups were different: 67.3 years and 60.1 years for losartan and amlodipine, respectively. The half-lives of amlodipine (30 to 60 h) and losartan and its carboxylic active metabolite (about 2 h and 6–9 h, respectively) were also different. The difference in age and in half-life of the drugs studied appeared to be serious limitations of the study for a fair comparison between the drugs.

Unfortunately, there is, as yet, no published study using randomized double-blind clinical study exploring one FIASMA against placebo with mortality as principal criterion.

## 4. Conclusions

The present review highlights the particular interest of the repurposing of FIASMAs for the inhibition of SARS-CoV-2 entry, and potential limitations of some published retrospective or prospective studies.

Future studies should pay better attention to the pharmacokinetic properties of the drugs investigated to allow the control of potential confounding factors found in retrospective studies; and to optimize the design of prospective controlled clinical trials (e.g., relevance of a loading dose for drugs with prolonged elimination half-lives). The present review highlights the particular interest of the repurposing of FIASMAs for the inhibition of SARS-CoV-2 entry. Very recent review highlights the interest of inhibition of ASM in SARS-CoV-2 infection, and suggests that serum level of shingosine-1-phosphate (S1P), one of the shingolipids, could be a prognostic factor for COVID-19 severity [104].

However, a recent research article has highlighted that phospholipidosis was a shared mechanism underlying the SARS-CoV-2 antiviral activity of many repurposed drugs without mentioning ASM inhibition [105].

Besides the pharmacokinetic properties, the choice of a FIASMA should consider the target population. Studies implemented in subjects just after a positive RT-PCR test should consider FIASMA with a low rate of undesirable adverse effects. Thus, antipsychotics (e.g., chlorpromazine) should not be considered and could be contraindicated. Studies implemented in subjects with a recent infection requiring hospitalization should ideally consider FIASMA combining other mechanisms of action to reduce excessive inflammatory response during sepsis or to attenuate neurological complications. For example, amitriptyline exerts neuroprotection via tropomyosin receptor kinase [106], and fluvoxamine via the Sigma-1 receptor activity may modulate SARS-CoV-2 induced cytokine storm [107].

## Figures and Tables

**Table 1 pharmaceuticals-14-00691-t001:** Functional inhibitors of acid sphingomyelinase (FIASMAs) with activity against SARS-CoV-2 on in silico, or/and in vitro models or/and in vivo models (N = 49).

FIASMAs	In Silico	(References)	In Vitro	(References)	In Vivo	(References)
Ambroxol					❖	[100]
Amiodarone	□□□	[15,19,26]	■■■	[12,17,18]	❖❖†	[20,103]
Amitriptyline	□□□□□□	[19,21,22,23,24,25]	■	[3]	❖†❖†	[54,103]
Amlodipine	□□□□□	[19,23,24,26,89]	■■■	[27,28,29]	❖❖❖❖†❖	[29,30,31,32,103]
Astemizole	□	[93]				
**Benztropine**			■■	[42,45]		
Bepridil	□□	[19,22]	■	[33]		
Carvedilol	□□□	[19,25,34]			❖❖	[25,53]
Cepharanthine	□□	[35,39]	■■■■■■	[36,37,38,40,41,94]		
**Chlorpromazine**	□□□□	[19,26,35,43]	■■■■■	[42,43,44,45,55]	❖❖❖†	[45,46,73]
Chlorprothixene	□	[19]	■	[18]		
Clemastine	□□□□	[19,35,47,48]	■■	[17,18]		
Clofazimine	□□	[19,91]	■■■■■■	[28,41,49,50,51,95]	❖	[51]
Clomiphene			■■	[52,96]		
**Clomipramine**	□	[19]	■■■■	[42,45,55,97]	❖†	[54]
Cloperastine	□□	[35,47]				
Cyclobenzaprine	□	[19]				
Cyproheptadine	□	[19]				
Desipramine	□	[19]	■■	[3,55]		
Desloratadine	□□	[19,48]	■	[56]		
Dicycloverine			■	[95]		
Dilazep	□	[57]				
Doxepine			■	[58]		
**Emetine**	□□□□□□	[35,59,65,66,67,90]	■■■■■■	[60,61,62,63,64,98]		
Flunarizine	□	[19]	■	[18]		
Fluoxetine	□□□□□	[19,57,69,70,71]	■■■■■■	[3,12,18,55,68,72]	❖❖†	[54,103]
**Fluphenazine**	□□	[19,74]	■■	[42,45]		
Flupenthixol	□	[70]	■■	[17,55]		
Fluvoxamine			■	[55]	❖❖	[75,76]
Hydroxyzine	□	[48]	■	[77]	❖❖❖†❖†	[77,78,101,103]
Imipramine	□□	[19,22]	■■■	[3,12,55]		
**Loperamide**			■■	[18,36]	❖†	[103]
Loratadine	□	[19]	■	[56]	❖❖	[77,88]
Maprotiline	□	[19]	■■	[3,18]		
Melatonine	□□□□□□	[25,26,34,79,80,81]			❖❖❖❖	[25,53,82,102]
Nortriptyline	□	[57]				
Paroxetine	□	[34]	■	[43]	❖❖❖†	[53,54,103]
Perphenazine	□	[19]				
Pimozide			■■	[33,55]		
Promazine	□	[19]				
**Promethazine**	□□□	[19,48,92]	■	[45]		
Protriptyline	□	[19]				
Quinacrine	□□□□	[34,48,71,83]	■■■■	[84,85,86,99]		
Sertraline			■■	[13,28]	❖†	[54]
**Tamoxifene**	□□	[48,92]	■■	[45,96]		
Thioridazine	□□	[71,87]	■	[28]		
Trifluoperazine	□□	[19,70]	■	[28]		
**Triflupromazine**	□	[19]				
Trimipramine	□□	[19,22]	■	[37]		

In bold: 9 drugs active against the 3 coronaviruses; in silico (□), in vitro (■), in vivo (❖) and negative result (†).

**Table 2 pharmaceuticals-14-00691-t002:** Functional inhibitors of acid sphingomyelinase (FIASMAs) (N = 15) with activity against SARS-CoV-2 in epidemiological, clinical studies, or case reports.

FIASMA (Reference)	Study Design	Sample Size	FIASMA Prevalence	Outcome
Amiodarone [20]	* Case report	1	100%	Case report of a 74-year-old man affected by respiratory failure related to COVID-19 who recovered after only supportive measures and amiodarone lasted 5 days.
[103]	* Retrospective	2602	1.27%	Mortality or intubation on hospitalized COVID-19 patients (N = 33 on amiodarone, N = 2569 without FIASMAs) HR = 1.26 (*p* = 0.14).
Amitriptyline [54]	* Retrospective	6924	0.56%	Mortality or intubation on hospitalized COVID-19 patients (N = 39 on amitriptyline, N = 6885 without antidepressants) HR = 0.85 (*p* = 0.59).
[103]	* Retrospective	2589	0.77%	Mortality or intubation on hospitalized COVID-19 patients (N = 20 on amitriptyline, N = 2569 without FIASMAs) HR = 0.54 (*p* = 0.12).
Amlodipine [29]	Retrospective	96	19.8%	Mortality on COVID-19 inpatients with hypertension as the only comorbidity. Patients on amlodipine (N = 19) or non-amlodipine (N = 77) had lower mortality (0% vs. 19.5%, *p* = 0.037).
[30]	Retrospective	65	36.9% ?	Mortality on elderly patients hospitalized for COVID-19; 24 were on amlodipine or nifedipine and 41 were not, 50% survived in the amlodipine or nifedipine group and 14.6% in the other group (*p* = 0.0036).
[31]	Retrospective	317	18.9%	Mortality on hospitalized COVID-19 patients; 60 were on amlodipine and 257 were not. Multiple logistic regression found lower mortality on patients on amlodipine (OR = 0.24, *p* = 0.0031).
[32]	* Prospective randomized	80	48.7%	Mortality. Losartan (N = 41) and amlodipine (N = 39) on patients with COVID-19 and primary hypertension. No significant difference of 30-day mortality rate.
[103]	* Retrospective	2666	3.64%	Mortality or intubation on hospitalized COVID-19 patients (N = 97 on amlodipine, N = 2569 without FIASMAs) HR = 0.7 (*p* = 0.037).
Carvedilol [25]	Retrospective	26,779	2.93%	PCR-positive. Patients tested for COVID-19 in Cleveland Clinic Health System; Carvedilol use (N = 785) was significantly associated with reduced likelihood of PCR positive to SARS-CoV-2 (OR = 0.74; *p* < 0.05).
[53]	Retrospective	11,672	2.96%	PCR-negative. Patients tested for COVID-19 in Cleveland Clinic Health System. Among 346 subjects on Carvedilol, 333 (96.2%) were PCR-negative and 13 (3.8%) were PCR-positive (*p* = 0.022).
Chlorpromazine [73]	Observational			Prevalence of COVID-19. Low rate (4%) of symptomatic COVID-19 infection in patients treated by antipsychotics than the rate (14%) observed in nurses or physicians in the same departments of psychiatry.
[46]	* Retrospective	14,340	0.38%	Mortality or intubation on hospitalized COVID-19 patients (N = 55 on chlorpromazine, N = 14,285 without chlorpromazine); 23.6% deaths on chlorpromazine and 9% deaths on subjects without chlorpromazine HR = 2.01 (*p* = 0.16).
Clomipramine [54]	* Retrospective	6894	0.13%	Mortality or intubation on hospitalized COVID-19 patients (N = 9 on clomipramine, N = 6885 without antidepressants) HR = 0.44 (*p* = 0.4).
Desloratadine [103]	* Retrospective	2576	0.27%	Mortality or intubation on hospitalized COVID-19 patients (N = 7 on desloratadine, N = 2569 without FIASMAs) HR = 0.68 (*p* = 0.44).
Fluoxetine [54]	* Retrospective	6915	0.43%	Mortality or intubation on hospitalized COVID-19 patients (N = 30 on fluoxetine, N = 6885 without antidepressants) HR = 0.37 (*p* = 0.049).
[103]	* Retrospective	2583	0.54%	Mortality or intubation on 2583 hospitalized COVID-19 patients (N = 14 on fluoxetine, N = 2569 without FIASMAs) HR = 0.3 (*p* = 0.082).
Fluvoxamine [75]	Double-blind randomized	152	52.6%	Clinical deterioration within 15 days. Fluvoxamine (N = 80) vs. placebo (N = 72) on non-hospitalized adults. Less clinical deterioration within 15 days of randomization in fluvoxamine group (0/80) than in placebo group (6 /72) (log-rank *p* = 0.009).
[76]	Prospective	113	57.5%	Incidence of hospitalization was 0% (0 of 65) with fluvoxamine and 12.5% (6 of 48) without fluvoxamine (*p* = 0.005).
Hydroxyzine [77]	Retrospective	219,000	0.12%	Incidence PCR-positive. Prior usage of hydroxyzine (N = 269) was associated with reduced incidence of positive SARS-CoV-2 in individuals 61 years and above.
[78]	* Retrospective	7345	1.88%	Mortality or intubation on hospitalized COVID-19 patients (N = 138) on hydroxyzine), (N = 7207) without hydroxyzine; HR = 0.42 (*p* = 0.001).
[101]	Retrospective	230,376	1.7%	Incidence PCR-negative. Prior usage of hydroxyzine (N = 3909) was not associated with increased incidence of negative SARS-CoV-2 in individuals. Adjusted OR = 0.76 (*p* > 0.05).
[103]	* Retrospective	2600	1.19%	Mortality or intubation on hospitalized COVID-19 patients (N = 31 on hydroxyzine, N = 2569 without FIASMAs) HR = 0.43 (*p* = 0.04).
Loperamide [103]	* Retrospective	2578	0.35%	Mortality or intubation on hospitalized COVID-19 patients (N = 9 on loperamide, N = 2569 without FIASMAs) HR = 0.25 (*p* = 0.1).
Loratadine [77]	Retrospective	219,000	0.13%	Incidence PCR-positive. Prior usage of loratadine (N = 284) was associated with reduced incidence of positive SARS-CoV-2 in individuals 61 years and above.
[88]	Case report	1	100%	Case report (54-year-old female) of pityriasis rosea gibert associated with COVID-19 infection hospitalized and treated with 200 mg/day hydrocortisone hemisuccinate and loratadine 20 mg/day. Two weeks after admission, the patient was discharged with a negative RT-PCR and without respiratory symptoms.
Melatonine [25]	Retrospective	26,779	3.94%	Incidence PCR-positive. Patients tested for COVID-19 in Cleveland Clinic Health System. Melatonine use (n = 1055) was significantly associated with reduced likelihood of PCR-positive to SARS-CoV-2 (OR = 0.72; *p* < 0.05).
[53]	Retrospective	11,672	4.53%	Incidence PCR-positive. Patients tested for COVID-19 in Cleveland Clinic Health System. Among 529 subjects on melatonin, 513 (97%) were PCR-negative and 16 (3%) were PCR-positive (*p* = 0.001).
[82]	Retrospective	791		Survival rate. Patients with COVID-19 infection. Melatonin exposure was associated with survival in COVID-19 patients.
[102]	Prospective longitudinal (before-after)	110	20%	Survival scores. Five groups of 22 patients were receiving pentoxifylline and one group had also 5 mg of melatonine every 12 h for 5 days. The medications improved the survival scores, and several inflammation markers (CRP…) were diminished at the end of the treatment
Paroxetine [53]	Retrospective	11,672		Incidence PCR-positive. Patients tested for COVID-19 in Cleveland Clinic Health System (7% PCR+). Among subjects on paroxetine, there was significant higher PCR-.
[54]	* Retrospective	6948	0.91%	Mortality or intubation in hospitalized COVID-19 patients (N = 63 on paroxetine, N = 6885 without antidepressants) HR = 0.52 (*p* = 0.0006).
[103]	* Retrospective	2610	1.57%	Mortality or intubation on 2610 hospitalized COVID-19 patients (N = 41 on paroxetine, N = 2569 without FIASMAs) HR = 0.66 (*p* = 0.13).
Sertraline [54]	* Retrospective	6907	0.32%	Mortality or intubation in hospitalized COVID-19 patients (N = 22 on sertraline N = 6885 without antidepressants) HR = 0.68 (*p* = 0.13).
[103]	* Retrospective	2590	0.81%	Mortality or intubation on 2590 hospitalized COVID-19 patients (N = 21 on sertraline, N = 2569 without FIASMAs) HR = 0.57 (*p* = 0.11).

***** Studies exploring acute and not chronic intake of FIASMAs asking the question of non-obtaining the steady state allowing a maximal ASM inhibition; HR: hazard ratio; OR: odds ratio.

## Data Availability

Data is contained within the article.

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
