# Peer review of "Update on Functional Inhibitors of Acid Sphingomyelinase (FIASMAs) in SARS-CoV-2 Infection"

_pharmaceuticals, 2021, doi:10.3390/ph14070691_

Round 1

Reviewer 1 Report

The perspective of Dr. Loas and La Colle concerns with the use of functional inhibitors of acid sphingomyelinase in SARS-CoV2 infections.

The perspective is well written. The authors provided a useful historical background about the FIASMA in the SARS-CoV2 infections and then all the updates on 16 June 2021. The Tables are well built and give a big help to the reader to have an immediate overview of the state of the art of FIASMA.

The manuscript can be accepted as it is

Author Response

No responses.

Reviewer 2 Report

The review covers the knowledge of the use of FIASMAs on SARS-CoV-2 infection. Though the authors covered most of the details, I have few comments to be addressed

1) In line 32, correct the word entry.

2) Please mention the reference for the sentence in line 34.

3) In Table 1, references are mentioned for Dilazep and Doxepine without mentioning the type of the study.

4) Data presented in both the tables is confusing, please try to represent the data in other way so that it is easily understood.

5) It would be good to mention details about the different types of Acid sphingomyelinases and the effect of drugs mentioned in the study on Acid sphingomyelinases.

6) Conclusions part should be rewritten, the present conclusions mentioned are pretty weak.

Author Response

The different points have been taken into account:

1) line 32: enter and not entry

2) line 34: reference is mentioned

3) In table 1: types of studies are mentioned for Dilazed and Doxepine

4) Table 2 has been extensively redesigned and is now easily understood

5) The two types of acid sphingomyelinase are mentioned lines 35-37 and 41

6) The conclusion has been rewritten and two new references discussed and mentioned in the references list.